# The Pharmacokinetic and Pharmacodynamic Relationship of Clinically Used Antiseizure Medications in the Maximal Electroshock Seizure Model in Rodents

**DOI:** 10.3390/ijms26157029

**Published:** 2025-07-22

**Authors:** Luis Bettio, Girish Bankar, Celine M. Dubé, Karen Nelkenbrecher, Maja Filipovic, Sarbjot Singh, Gina DeBoer, Stephanie Lee, Andrea Lindgren, Luis Sojo, Richard Dean, James P. Johnson, Nina Weishaupt

**Affiliations:** Xenon Pharmaceuticals Inc., 3650 Gilmore Way, Burnaby, BC V5G 4W8, Canada

**Keywords:** antiseizure medications, brain-to-plasma ratio, concentration–response, epilepsy, maximal electroshock seizure, pharmacokinetics, pharmacodynamics

## Abstract

The assessment of the efficacy of antiseizure medications (ASMs) in animal models of acute seizures has played a critical role in these drugs’ success in clinical trials for human epilepsy. One of the most widely used animal models for this purpose is the maximal electroshock seizure (MES) model. While there are numerous published reports on the efficacy of conventional ASMs in MES models, there is a need to expand the understanding on the brain concentrations that are needed to achieve optimal levels of efficacy in this model. We assessed the pharmacokinetic/pharmacodynamic (PK/PD) profiles of six ASMs, namely carbamazepine (CBZ), phenytoin (PHT), valproic acid (VPA), lacosamide (LSM), cenobamate (CNB), and retigabine (RTG), using MES models in mice and rats. EC_50_ values for plasma and the brain were generally higher in mice than rats, with fold differences ranging from 1.3- to 8.6-fold for plasma and from 1.2- to 11.5-fold for brain. Phenytoin showed the largest interspecies divergence. These results suggest that rats may exhibit greater sensitivity to seizure protection in the MES model, likely reflecting species differences in metabolism and brain penetration. These findings highlight the value of considering concentration–response variations and species-specific differences when assessing the efficacy of both conventional ASMs and novel compounds exhibiting anticonvulsant activity.

## 1. Introduction

The maximal electroshock seizure (MES) model is one of the most widely used experimental paradigms to evaluate the efficacy of compounds against tonic–clonic seizures [1]. This test induces synchronous neural discharges in the brain through the application of electrical current using transcorneal or transauricular electrodes [2]. One of the main advantages of an MES model is its ability to quickly and reliably induce tonic–clonic seizures, allowing for the rapid assessment of potential antiseizure medications (ASMs). Standard and newly developed ASMs tend to prevent epileptic activity induced by an MES procedure through mechanisms that increase the threshold for seizure initiation and prevent its spread [3]. Despite ongoing efforts to develop new models that closely mimic human epilepsy, the MES model remains the most reliable and effective tool for identifying anticonvulsant compounds.

Commonly used in clinical practice, conventional ASMs such as carbamazepine (CBZ), phenytoin (PHT), and valproic acid (VPA) are still being studied in preclinical settings, particularly for their interactions with other drugs in the MES model [4,5,6]. Nevertheless, there is a need to expand the understanding of the pharmacokinetic/pharmacodynamic (PK/PD) profiles of different ASMs in rodents in this experimental paradigm, especially in terms of exploring interspecies variations. The choice between mice and rats is one factor that can influence experimental outcomes when assessing the effectiveness of new ASMs. Both species have been extensively used in the evaluation of novel drug candidates for epilepsy, and compounds that demonstrate efficacy in a mouse MES typically produce similar effects in a rat MES [7]. However, metabolic differences and variations in tissue distribution between both species can lead to distinct pharmacokinetic profiles, which can affect the potency of the compounds tested [8,9]. Notably, there is a shortage of studies comparing the PK/PD profiles of different ASMs in both mice and rats.

In addition, few studies report concentrations of ASMs in the brain and plasma, which is useful information to understand a drug’s PK/PD profile. The ability of a drug to cross the blood–brain barrier (BBB) and reach its target site in the brain is expressed by the brain-to-plasma ratio (B/P ratio): the concentration of a drug in the brain relative to its concentration in the plasma [10]. Drugs with a high B/P ratio are more likely to penetrate the BBB and present low P-glycoprotein (Pgp) efflux transport [11]. However, despite the close relationship between the efficacy of drugs that act in the central nervous system (CNS) and their B/P ratio, the variation in this parameter among different rodent species has not been studied in great detail [12]. Increased understanding of the B/P ratio would allow for a better comparison of preclinical data as it relates to the effectiveness and safety of ASMs.

In-depth understanding of the interactive dynamics of PK/PD processes in a rodent model like the MES may provide valuable insights for pre-clinical drug discovery. Here, we conducted a comprehensive evaluation of five commonly prescribed ASMs, namely CBZ, PHT, VPA, lacosamide (LSM), and cenobamate (CNB), in mice and rats using an MES model. Retigabine (RTG), although no longer marketed, was included due to its distinct mechanism of action as a Kv_7_ channel opener. We determined the half-maximal effective concentration (EC_50_) of each compound in the plasma and brain and their B/P ratios, providing significant insights into the variations in the PK/PD profiles of both species. 

## 2. Results

The anticonvulsant activity of CBZ (a sodium channel blocker) against MES-induced seizures is shown in Figure 1A,B. Mice treated with CBZ produced a significant reduction in fraction seizing at doses ranging from 6 to 30 mg/kg (*p* < 0.0001) with a median effective dose (ED_50_) of 9.67 mg/kg. In rats, MES-induced seizures were reduced at doses from 3 to 30 mg/kg (*p* < 0.0001) with an ED_50_ of 4.39 mg/kg. The EC_50_ of CBZ in mouse plasma (13.8 µM) was about three times higher than the EC_50_ in rat plasma (4.55 µM). The brain EC_50_ was nearly seven times higher in mice compared to rats (EC_50_ mouse = 32.5 µM and EC_50_ rat = 4.70 µM) (Figure 2A–C).

The sodium channel blocker PHT induced a dose-dependent effect in mice at doses ranging from 9 to 18 mg/kg (*p* < 0.0001, ED_50_ = 9.81 mg/kg) and in rats at doses from 10 to 75 mg/kg (*p* < 0.0001, ED_50_ = 16.9 mg/kg) (Figure 1C,D). The plasma EC_50_ in mice treated with PHT (27.6 µM) was about nine times higher than in rats (3.20 µM). This difference was even more pronounced in the brain, where the EC_50_ in mice (30.6 µM) was about 12 times higher than in rats (2.66 µM) (Figure 2D–F).

The mechanisms underlying the antiseizure activity of VPA are not fully understood, but it is known to increase γ-aminobutyric acid (GABA)-mediated signaling and block sodium channels [13]. In our study, VPA reduced seizures induced by MES in mice at doses ranging from 200 to 300 mg/kg (*p* < 0.0001, ED_50_ = 196 mg/kg) and in rats at doses ranging from 375 to 750 mg/kg (*p* < 0.0001, ED_50_ = 366 mg/kg) (Figure 1E,F). The plasma EC_50_ values were similar in mice (1668 µM) and rats (1306 µM). In the brain, however, the EC_50_ was twice as high in mice (630 µM) compared to rats (307 µM) (Figure 3A–C).

The antiepileptic activity of LSM is thought to be due to its ability to enhance the inactivation of voltage-gated sodium channels and interact with collapsin response mediator protein-2 (CRMP2) [14]. In our study, LSM was effective against MES-induced seizures in mice at doses ranging from 10 to 75 mg/kg (*p* < 0.0001, ED_50_ = 15.2 mg/kg) and in rats at doses from 10 to 25 mg/kg (*p* < 0.0001, ED_50_ = 9.80 mg/kg) (Figure 1G,H). Mice treated with LSM exhibited similar EC_50_ values in both the plasma (19.3 µM; 0.78-fold difference) and the brain (14.8 µM; 1.21-fold difference) as rats (EC_50_, plasma: 24.9 µM; EC_50_, brain: 12.2 µM) (Figure 3D–F).

The mechanism of CNB remains unclear, but it has been proposed that it exert antiepileptic activity through a dual mechanism of action. This drug has been suggested to act as a voltage-gated sodium channel blocker as well as an enhancer of GABAergic activity [15]. In our study, CNB reduced MES-induced seizures in mice at doses of 7.5, 10, and 30 mg/kg (*p* < 0.0001, ED_50_ = 7.05 mg/kg) and at the dose of 18 mg/kg in rats (*p* < 0.0001, ED_50_ = 6 mg/kg) (Figure 1I–J). We observed that mice had a 4-fold higher EC_50_ in plasma (66.0 µM) and a 2-fold higher EC_50_ in the brain (24.9 µM) in comparison to rats (15.9 µM and 10.7 µM, respectively) (Figure 4A–C).

We also examined the effectiveness and tissue distribution of RTG, a KV_7_ potassium channel potentiator that has shown clinical efficacy in reducing seizures [16]. We observed a dose–response curve at doses ranging from 5 to 35 mg/kg; however, despite finding a 50% reduction in seizure score compared to the vehicle-treated group, the results did not reach statistical significance (*p* = 0.12, ED_50_ = 42.9 mg/kg). In rats, a significant reduction in seizure frequency was observed at 30 mg/kg (*p* < 0.01, ED_50_ = 13.9 mg/kg) (Figure 1K,L). Mice EC_50_ values were 1.5-fold higher in plasma (2.26 µM) and 1.19-fold higher in the brain (3.58 µM) compared to rats (1.46 µM and 3.01 µM, respectively) (Figure 4D–F).

The results described above are summarized in Table 1, which also provides the B/P values for each experimental group. Overall, brain and plasma EC_50_ values were higher in mice than in rats for the ASMs tested (Figure 5). Mice exhibited higher brain penetrance for CBZ, PHT, VPA, and LSM compared to rats (2.28-, 1.33-, 1.61-, and 1.57-fold, respectively) but showed lower brain penetrance for CNB and RTG (0.56- and 0.77-fold, respectively).

## 3. Discussion

Epilepsy is a serious neurological condition that is often accompanied by psychiatric comorbidities [17] and can lead to significant mortality [18]. Despite the availability of many ASMs, there are still significant challenges in the management of epilepsy. A major problem is the high proportion of drug-resistant cases, that is, when a patient does not respond to at least two tolerated ASMs used either in combination or separately. It is estimated that drug-resistant epilepsy, which requires adjunctive treatment, accounts for around 30% of cases [19,20]. While ASM combination therapy can increase the likelihood of achieving seizure control, it is also associated with an increased risk of adverse effects, which can lead to poor adherence to medication [21,22].

The variability in the PK/PD of ASMs in individual patients can make it difficult to predict the optimal therapeutic dose of an ASM, leading to a trial-and-error approach that can be time-consuming [23,24,25]. In some cases, pre-clinical studies providing additional information about the PK/PD profile of an ASM may be useful to improve dosing accuracy and reduce the time required for achieving adequate seizure control and minimizing side effects. In addition, an improved understanding of the relationship between the efficacy of an ASM and brain exposure might be obtained from data linking plasma levels of ASMs to anticonvulsant activity in pre-clinical studies. In this study, we evaluated the PK/PD profile and B/P ratio of ASMs using MES models, a widely used paradigm to investigate anticonvulsant drug activity that is thought to be able to reasonably estimate human efficacious doses and exposure [1]. The efficacy and exposure levels of six ASMs with established clinical PK and efficacy data were evaluated in an MES model using CF-1 mice and Sprague-Dawley rats. As with any preclinical research, heterogeneity in the EC_50_ values can be expected between studies due to strain-specific differences and methodological variations [26,27].

Our study found that CBZ, PHT, VPA, LSM, CNB, and RTG all had substantial anticonvulsant activity in the tested MES model. Although RTG did not produce a statistically significant anticonvulsant effect in mice at the doses tested (5–35 mg/kg), we observed a 50% reduction in fraction seizing at the highest dose tested. Since mice treated with RTG at higher doses began to show clinical signs, we did not test additional doses to expand the dose–response curve.

Rodent ED_50_ from pre-clinical dose–response studies is often used as an early predictor of antiseizure effectiveness in humans [1]. Alternatively or in addition, determining the EC_50_ of a drug can provide valuable insights that cannot be obtained from ED_50_, given that the concentration of a drug in the blood and tissues is often a more accurate predictor of its pharmacological effects than the dose [28], as it can provide more information about the individual variability in pharmacological processes such as absorption, distribution, metabolism, and elimination [29]. The majority of published studies evaluating ASMs exclusively present ED_50_ values, and those where concentration is evaluated often assess only plasma EC_50_ values. Our study expands upon current knowledge by providing brain EC_50_ values for the assessed ASMs, providing a thorough evaluation of the PK/PD profiles of these CNS-targeting ASMs. Further, while most studies evaluate ASMs using mice or rats, there are few systematic comparisons regarding drug efficacy between these species. Establishing these comparisons could allow for a more comprehensive understanding of species-specific responses, which may refine the reliability and translatability of the findings in preclinical research.

The results of our EC_50_ analyses in the plasma and brain of rodents indicate that the sodium channel blockers CBZ, PHT, and CNB were several times more potent in rats compared to mice. However, no interspecies difference was observed in rodents treated with LSM. Moreover, we found that rats and mice that received the GABAergic modulator VPA had similar EC_50_ in plasma, but the EC_50_ in the brain of rats was about two times lower than the EC_50_ in mice. VPA has previously been reported to present a wide range of half-lives in different species, with a shorter half-life in rats compared to mice [8]. This difference may be due to species-specific differences in drug-metabolizing enzymes [8] and/or drug-specific interactions with BBB permeability and/or efflux transporters [30]. In addition to evaluating conventional ASMs, we investigated the PK/PD of RTG, an ASM with a different mechanism of action (KV_7_ potassium channel potentiator) compared to the other ASMs tested. We observed that RTG was equipotent in mice and rats. 

While our study provides EC_50_ and B/P ratio estimates at a fixed time point, we acknowledge that these values may not reflect full distributional equilibrium, particularly for compounds with delayed brain penetration [8,12,31]. With this limitation in mind, Appendix A provides some context to how results from MES studies in mice and rats—which are commonly used to evaluate the antiseizure efficacy of preclinical drug candidates—relate to reported clinical data, taking plasma protein binding into account. Future studies incorporating compartmental PK/PD modeling with time-course sampling would allow for more precise estimation of dynamic exposure–response relationships and brain–plasma partitioning. 

In conclusion, our study provides a comparative analysis of the PK/PD profiles of six ASMs in rodent MES models. We found that conventional ASMs were generally more effective in reducing seizures in rats than in mice, underscoring the importance of considering interspecies differences in drug metabolism, brain penetrance, and protein binding when interpreting preclinical efficacy data. One limitation of our analysis is that it was based on retrospective in-house data and was not designed to comprehensively evaluate all ASM classes. The inclusion of additional compounds such as lamotrigine may have provided broader mechanistic insight. Moreover, the number of dose groups per compound was limited by dataset availability, introducing variability in curve-fitting precision (particularly for VPA and PHT). These limitations should be considered when interpreting exposure–response relationships. Future studies may help clarify how physiological differences between rodents contribute to interspecies variation in drug exposure and efficacy, as well as how these factors can be integrated with other translational strategies to improve the predictive value of preclinical models.

## 4. Materials and Methods

### 4.1. Animals

CF-1 male mice (3–7 weeks old) and Sprague-Dawley rats (5 weeks old) of both sexes were used in the experiments (obtained from Charles River Laboratories). Animals were group-housed (2–4 mice or 3–4 rats per cage) in plastic cages and maintained under controlled temperature (18–25 °C), humidity, ventilation, and lighting (12 h light/dark cycle). They had free access to water and Certified Rodent Chow (Teklad Rodent Chow). Upon arrival, the animals were examined to ensure good health status and were acclimated to the facility for at least 5 days before being subjected to the MES protocol. All experiments were conducted in accordance with guidelines set by the Canadian Council on Animal Care and approved by the Xenon Animal Care Committee.

### 4.2. Drugs and Treatments

To study the PK/PD profiles of conventional ASMs, the following drugs were investigated: CBZ (Sigma Aldrich C4024, Milwaukee, WI, USA); PHT (Sigma Aldrich, Laramie, US); VPA (Sigma Aldrich P4543, Milwaukee, WI, USA); LSM (Toronto Research Chemicals L098500, Vaughan, ON, Canada); CNB (Med Chem Express HY17607); and RTG (synthesized at Xenon Pharmaceuticals Inc., Burnaby, BC, Canada). Drugs were administered by the intraperitoneal route (IP; dissolved in sterile saline) or oral route (PO; dissolved in 0.5% methylcellulose + 0.2% Tween 80 in DI water) in a volume of 10 mL/kg. Control animals received the appropriate vehicle. The doses and pre-treatment times for each drug, detailed in Table 2, were determined based on in-house data or, in some cases, from the publicly available PANACHE database.

### 4.3. Alternating Current-Maximal Electroshock Seizure Assay (AC-MES)

The MES assay is widely used in the search for anticonvulsant substances [26,32,33]. This animal model for generalized tonic–clonic seizures assesses a compound’s ability to prevent seizure spread. In our studies, an alternating current electroshock (60 Hz, 40 mA for mice and 150 mA for rats) was delivered for 0.2 s (0.5 ms pulse width) using corneal electrodes (HSE-HA Rodent Shocker, Harvard Apparatus, model 73-0105). Before the electroshock stimulation, the eyes were anesthetized with a topical application of 0.5% proparacaine hydrochloride (one drop per eye). Animals were then restrained; the corneal electrodes were applied, and the shock was administered. An MES-induced seizure was characterized by an initial generalized tonic seizure with a hindlimb tonic extensor component.

In each experiment, animals were randomly assigned to a treatment group, ensuring that all animals had an equal and unbiased chance of being assigned to any treatment group. The protocols for administering PO or IP doses of the vehicle or drug are described in Table 1. These doses were administered 0.5–2 h before the application of electric stimulation. An animal was considered protected from MES-induced seizures if the hindlimb tonic extensor component of the seizure was abolished, and it was scored as “0”. If a mouse or rat displayed tonic hindlimb extension within 5 s post-stimulation, the score was “1”. Seizure induction and scoring were performed by an experimenter who was blinded to treatment conditions.

### 4.4. Collection of Plasma and Brain Samples

After being subjected to the stimulation protocol, rodents were anesthetized by isoflurane inhalation until they reached a surgical plane of anesthesia. Approximately 0.5 mL of blood was collected via cardiac puncture and transferred to EDTA-coated tubes on ice. Animals were then euthanized by cervical dislocation, and their brains were removed, weighed, and immediately frozen on dry ice. At the end of sample collection, blood was centrifuged at 4000 rpm for 10 min at 4 °C, and the plasma was separated. All samples were stored at −80 °C until the time of bioanalysis.

### 4.5. Processing of Samples for Bioanalysis

#### 4.5.1. Plasma Samples

Plasma samples were extracted for analysis using a protein precipitation method with acetonitrile. A total of 50 µL of diluted plasma samples was mixed with 50 µL of an internal standard (IS) solution in a 1:1 mixture of acetonitrile and water. A total of 200 µL of acetonitrile was then added to the mixture, which was vortexed for 30 s. The samples were centrifuged at 13,000 rpm for 20 min, decanted into a 96-well plate, and further centrifuged at 4000 rpm for 20 min. The samples were then analyzed using ultra-high-performance electrospray ionization tandem mass spectrometry (UHPLC-ESI-MS/MS).

#### 4.5.2. Brain Samples

Brains were homogenized in a 1:1 mixture of acetonitrile and water (2 mL per mouse brain) using the T18 ULTRA-TURRAX (IKA, Staufen im Breisgau, Germany) homogenizer for about 1 min. The homogenate was centrifuged at 13,000 rpm for 20 min, and 50 µL of the supernatant was treated in the same way as the plasma samples for analysis.

### 4.6. Bioanalysis Procedures

Dipotassium EDTA blank mouse plasma (Valley Biomedical, Winchester, VA, USA) was used to prepare standards and quality control (QC) samples for plasma quantitation and as surrogates for brain homogenate quantitation. Twelve calibration samples ranged from 2.34 ng/mL to 4800 ng/mL. QC samples at 14 ng/mL (QC-L), 225 ng/mL (QC-M), and 3600 ng/mL (QC-H) were analyzed in triplicate.

Samples were analyzed by UHPLC-ESI MS/MS using a TQ-5500 Sciex triple quadrupole mass spectrometer (AB Sciex, Concord, ON, Canada) equipped with a Shimadzu Nexera UHPLC pump and an auto-sampler system using an ACE C18 PFP, 2.50 × 50 mm, 1.7 µm particle size column and gradient elution consisting of solvent A (0.1% formic acid in water) and solvent B (0.1% formic acid in acetonitrile) starting at 50% B from 0 min to 0.6 min and then increased to 100% B from 0.6 min to 1.0 min. At 1.5 min, the mobile phase composition was switched back to 50% B at 1.6 min and then held at 50% B until 2.5 min. The flow rate used throughout the experiment was 0.4 mL/min. All compounds except phenytoin were detected in the positive ion mode using the mass transitions 237→194 *m*/*z* (CBZ), 145→71 *m*/*z* (VPA), 251.2→91 *m*/*z* (LSM), 267.8→155.1 *m*/*z* (CNB), and 304.2→109 *m*/*z* (RTG). PHT was detected in the negative ion mode using the mass transition 251→42 *m*/*z*. Each compound was matched with a proprietary co-eluting internal standard. The UHPLC-ESI MS/MS system was controlled, and samples were quantified using Analyst 1.6.

Sample concentrations were determined using a linear calibration function, weighted 1/x, and generated by the regression of analyte to IS peak area ratios in the standard samples to their respective concentrations. Acceptance criteria for the analytical run required the back-calculated values of the standards and the QC samples to fall within ±20% of their nominal values, except for the lowest standard or lower limit of quantitation, for which the acceptance criterion was ±25%. At least six out of twelve standard points had to show back-calculated values within ±20% of their nominal concentrations for the calibration to be accepted. At least three QC samples, one at each concentration, had to show back-calculated values within ±20% of their nominal concentrations for the whole sample batch to be valid.

### 4.7. Data Processing and Analysis

The statistical analysis for the data collected in this study was conducted using the GraphPad Prism version 9 software. The result of the MES test is presented as the fraction of animals that showed hindlimb extension (“Fraction Seizing”) out of the total number of animals in each group.

The AC-MES assay has a binary readout: animals are either protected (0) or experience seizures (1). Therefore, a logistic regression analysis was performed using individual animal response data to estimate the concentration where there is a 50% probability that the rodents will be protected from seizure, reported as EC_50_. While the raw data were binary, the resulting curve was fit to predict probabilities ranging from 0 to 1. Confidence intervals for EC_50_ were not derived due to the nature of the probability-based curve fitting. Model quality was assessed using standard logistic regression diagnostics, including the log-likelihood ratio and pseudo-R^2^ values (Tjur’s and Cox–Snell’s), which are reported in Appendix A. For data visualization purposes, the curve fit was overlayed against the concentration response data points plotted by dose.

For dose–response curves, data were plotted as mean seizure protection (%) by dose group, and curves were fitted using the Hill–Langmuir equation:Y = B + (T − B) × x^n^/(IC_50_^n^ + x^n^), 
where

B = bottom and is set as 0.T = top and is set as 1.n = the Hill coefficient and is constrained to less than zero.IC_50_ = the concentration of a compound required for 50% inhibition in vitro.

All experimental data is expressed as the mean, and between-group differences were analyzed using the Kruskal–Wallis test followed by Dunn’s multiple comparison test. Statistical significance was reached at *p* < 0.05.

Rodent PPB values were derived from Xenon Pharmaceutical’s in-house data, except for VPA [34] and CNB (mouse) [35], for which published data were used. PPB data is available in Appendix A.

## Figures and Tables

**Figure 1 ijms-26-07029-f001:**
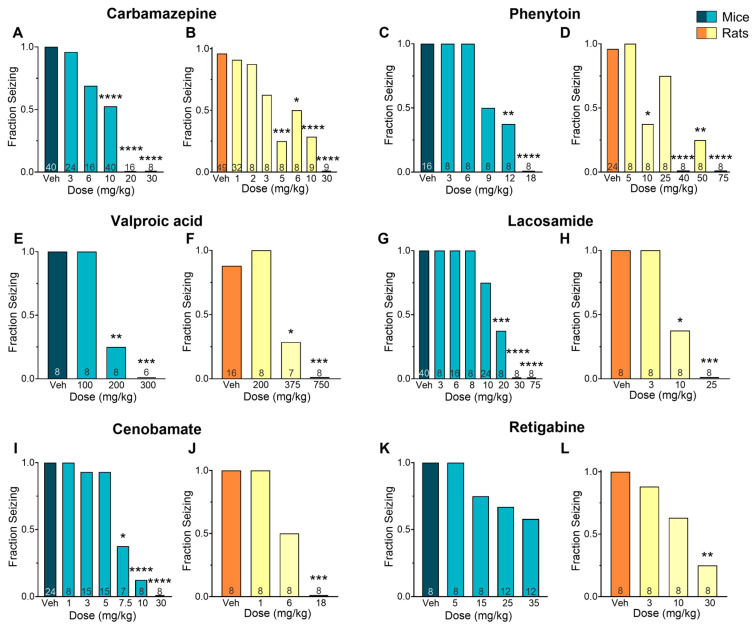
Efficacy of various antiseizure medications (ASMs) in mice and rats subjected to a maximal electroshock seizure (MES) test. The results show a clear dose–response relationship for carbamazepine (**A**,**B**), phenytoin (**C**,**D**), valproic acid (**E**,**F**), lacosamide (**G**,**H**), cenobamate (**I**,**J**), and retigabine (**K**,**L**). Data is presented as the mean. * *p* < 0.05; ** *p* < 0.01; *** *p* < 0.001; **** *p* < 0.0001 in comparison to vehicle-treated group (Kruskal–Wallis nonparametric ANOVA followed by Dunn’s multiple comparisons test). Numbers inside bars indicate the number of animals per group.

**Figure 2 ijms-26-07029-f002:**
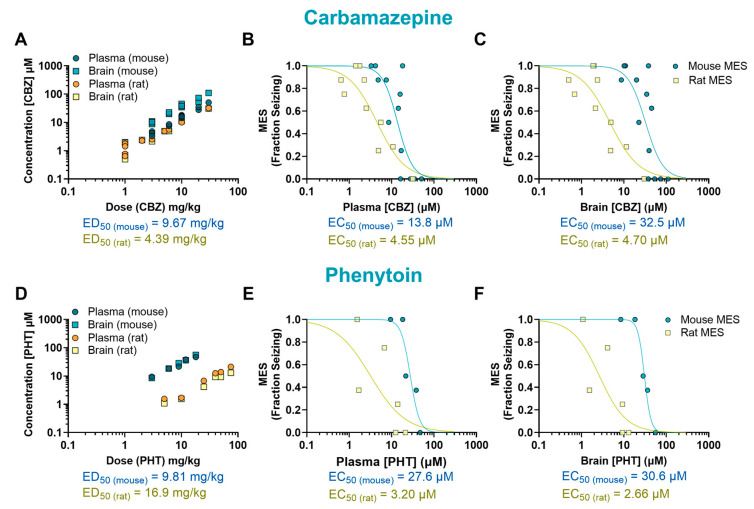
Pharmacokinetic/pharmacodynamic (PK/PD) profile of carbamazepine (CBZ) and phenytoin (PHT) in rodents exposed to a maximal electroshock seizure (MES) test. Relationship between dose and concentration in the plasma and brain of treated animals (**A**,**D**). Concentration–response of CBZ and PHT in the plasma (**B**,**E**) and brain (**C**,**F**) of animals that received a single oral dose before being tested in the seizure assay. Each point represents the average value from a given experimental group. Some of the doses have multiple data points as they were tested more than once (*n* = 8–9 per data point). EC_50_: half-maximal effective concentration; ED_50_: median effective dose.

**Figure 3 ijms-26-07029-f003:**
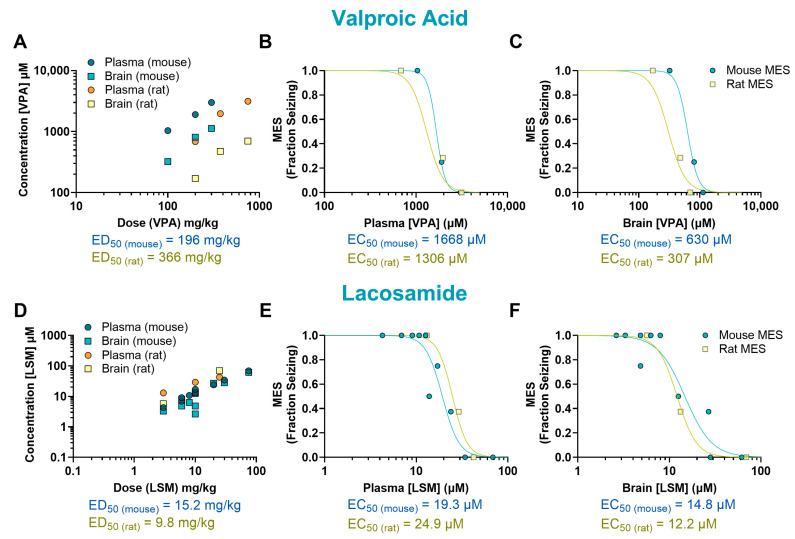
Pharmacokinetic/pharmacodynamic (PK/PD) profile of valproic acid (VPA) and lacosamide (LSM) in rodents exposed to a maximal electroshock seizure (MES) test. Relationship between the dosage administered and the resulting concentration in the plasma and brain of treated animals (**A**,**D**). Concentration–response relationship of VPA and LSM in the plasma (**B**,**E**) and brain (**C**,**F**) of animals that received a single oral dose before being tested in the seizure assay. Each point represents the average value from a given experimental group. Some of the doses have multiple data points as they were tested more than once (*n* = 6–8 per data point). EC_50_: half-maximal effective concentration; ED_50_: median effective dose.

**Figure 4 ijms-26-07029-f004:**
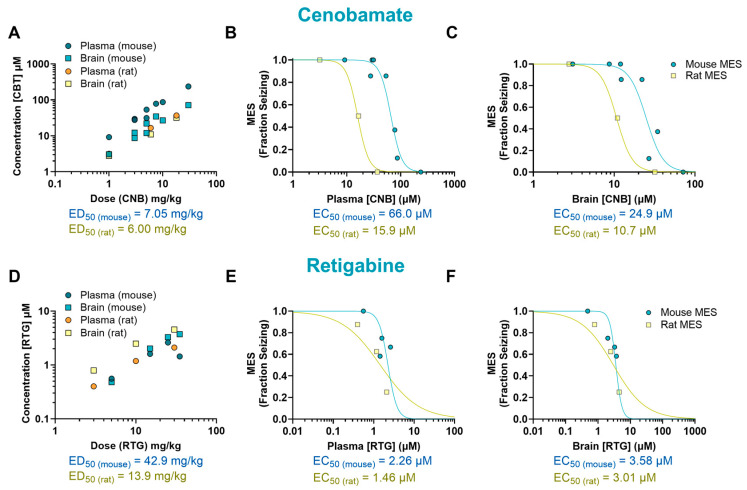
Pharmacokinetic/pharmacodynamic (PK/PD) profile of cenobamate (CNB) and retigabine (RTG) in rodents exposed to a maximal electroshock seizure (MES) test. Relationship between dose and concentration in the plasma and brain of treated animals (**A**,**D**). Concentration–response curve of CNB and RTG in the plasma (**B**,**E**) and brain (**C**,**F**) of animals that received a single oral dose before being tested in the seizure assay. Some of the doses have multiple data points as they were tested more than once (*n* = 7–12 per data point). EC_50_: half-maximal effective concentration; ED_50_: median effective dose.

**Figure 5 ijms-26-07029-f005:**
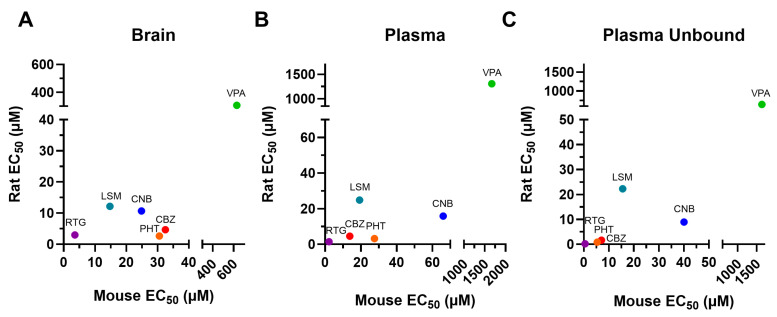
EC_50_ values for the brain (**A**), plasma (**B**), and unbound plasma (**C**) of rodents administered with a single dose of conventional antiseizure medications (ASMs). In general, the tested compounds exhibited greater potency in the rat MES compared to the mouse MES. CNB: cenobamate; CBZ: carbamazepine; EC_50_: half-maximal effective concentration; LSM: lacosamide; PHT: phenytoin; RTG: retigabine; VPA: valproic acid.

**Table 1 ijms-26-07029-t001:** Potency of conventional ASMs in the rodent MES.

Drug	Mechanism of Action	Species	ED_50_ (mg/kg)	EC_50_ (µM) Plasma	EC_50_ (µM) Brain	B/P Ratio
CBZ	Sodium channel blockade	Mouse	9.67	13.8	32.5	2.36
		Rat	4.39	4.55	4.70	1.03
		Mouse/rat fold	2.20	3.03	6.91	2.28
PHT	Sodium channel blockade	Mouse	9.81	27.6	30.6	1.11
		Rat	16.9	3.20	2.66	0.83
		Mouse/rat fold	0.58	8.63	11.5	1.33
VPA	GABA potentiation	Mouse	196	1668	630	0.38
		Rat	366	1306	307	0.24
		Mouse/rat fold	0.54	1.28	2.05	1.61
LSM	Sodium channel blockade/CRMP2 binder	Mouse	15.2	19.3	14.8	0.77
		Rat	9.80	24.9	12.2	0.49
		Mouse/rat fold	1.55	0.78	1.21	1.57
CNB	Sodium channel blockade/GABA potentiation suggested	Mouse	7.05	66.0	24.9	0.38
		Rat	6.00	15.9	10.7	0.67
		Mouse/rat fold	1.18	4.15	2.33	0.56
RTG	Kv_7_ potassium channel potentiator	Mouse	42.9	2.26	3.58	1.58
		Rat	13.9	1.46	3.01	2.06
		Mouse/rat fold	3.09	1.55	1.19	0.77

Note: ASMs: antiseizure medications; B/P: brain-to-plasma; CNB: cenobamate; CBZ: carbamazepine; EC_50_: half-maximal effective concentration; ED_50_: median effective dose; LSM: lacosamide; MES: maximal electroshock seizure; PHT: phenytoin; RTG: retigabine; VPA: valproic acid.

**Table 2 ijms-26-07029-t002:** Dosing times and routes of administration used to investigate the PK/PD profile of ASMs.

Drug	Mouse MES		Rat MES	
	Route	Pre-Treatment Time (h)	Route	Pre-Treatment Time (h)
CBZ	PO	0.5	PO	2
PHT	IP	1	PO	2
VPA	IP	0.25	PO	1
LSM	PO	2	PO	2
CNB	PO	2	PO	2
RTG	PO	0.5	PO	0.5

Note: ASMs: antiseizure medications; CNB: cenobamate; CBZ: carbamazepine; IP: intraperitoneal; LSM: lacosamide; MES: maximal electroshock seizure; PHT: phenytoin; PK/PD: pharmacokinetic/pharmacodynamics; PO: per os; RTG: retigabine; SD: Sprague-Dawley; VPA: valproic acid.

## Data Availability

Data is available upon request from the authors.

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
