# Peer review of "The Pharmacokinetic and Pharmacodynamic Relationship of Clinically Used Antiseizure Medications in the Maximal Electroshock Seizure Model in Rodents"

_ijms, 2025, doi:10.3390/ijms26157029_

Round 1

Reviewer 1 Report

Comments and Suggestions for Authors

The authors of the manuscript performed pharmacokinetic and pharmacodynamic analysis for carbamazepine, phenytoin, valproic acid, lacosamide, cenobamate, and retigabine for the induced seizure. Below are my comments.

1. The authors provided the 50% effective concentrations (i.e., EC50) of carbamazepine, phenytoin (Figure 2, and lines 81-82 and 93-95), valproic acid,  lacosamide (Figure 3, and lines 110-111 and 117-119), cenobamate, and retigabine (Figure 4, and lines 133-135 and 142-143). However, the dosing methods are per oral or intraperitoneal injection (Table 2), which does not make an equilibrium plateau of drug concentrations (e.g., IV infusion). In other words, the drugs' concentration should be different across time because of what is the primary interest of pharmacokinetics (i.e., quantitative rate and extent of absorption, distribution, metabolism and excretion). Additionally, the brain-to-plasma concentration ratio can vary over time because of the rate of distribution. Especially the brain is usually assumed to be a slow organ, which means the distribution will often delayed. The single-point measure of the brain-to-plasma concentration ratio cannot replace the brain-to-plasma partition coefficient. Please consider developing (or citing the literature models) a pharmacokinetic (e.g., traditional compartment models) and pharmacodynamic model (e.g., indirect response models) for each molecule to analyze your experimental data (e.g., estimating model-based brain-to-plasma partition coefficient, model-based EC50, and model-based response-exposure analysis).

2. The authors mentioned "exploring interspecies variations" in line 47 and human therapy (e.g., line 258, Figure 6). However, I think the above makes the discussion on the response-concentration relationship in this manuscript difficult to interpret for human therapy, which would be the ultimate goal of the research in pharmacy (e.g., exploring "interspecies variations" in line 47). Please consider performing species scaling based on various methods (e.g., allometric scaling, PBPK modelling, and so on) for the pharmacokinetic models (at least) and compare the unbound exposure of the molecules to discuss the human therapy.

Author Response

Please see attachment. Changes to the manuscript are highlighted yellow in the revised version.

Reviewer 2 Report

Comments and Suggestions for Authors

Thank you for the opportunity to review the paper by Bettio et al. This manuscript investigates the dose-response relationships of six anticonvulsants: carbamazepine (CBZ), phenytoin (PHT), valproic acid (VPA), lacosamide (LSM), cenobamate (CNB), and retigabine (RTG), in rats and mice. The animals were subjected to a maximal electroshock simulation model. Drug concentrations were measured in plasma and brain homogenates to find a direct relationship between drug exposure and the pharmacodynamic effects. Overall, the paper is well-written, but the research described is not entirely convincing.

Major concerns

  1. It is not clear why the authors chose these drugs for the investigation. Concerning the diversity of the mechanisms through which antiepileptic drugs can exert their effects, the study would be more compelling if one could compare the EC50 values of substances clearly representing these differences. Alternatively, the investigation could be intended to be comprehensive at least in terms of the anticonvulsants employed massively. In this case, at least lamotrigine should also be added. It is not obvious why retigabine was included even though it was withdrawn from markets long time ago.
  2. Sample size has not been provided, yet one can infer from the plots in Figure 4 that it was different for the various drugs. There are very few data points for valproate and phenytoin, casting doubt on the credibility of the sigmoids fitted. The sample size and the reason for differences should be clarified. The removal of data on valproate and phenytoin should be considered.
  3. The study design lacks the in-depth investigation of the reasons behind the similarities and differences described. The conclusion that interspecies differences can be so enormous and inconsistent warrants compelling evidence, and also the clarification of underlying biology.
  4. The authors compare the results to human „therapeutic plasma concentration ranges”. In view of the emergence of precision pharmacotherapy, the light evidence substantiating these therapeutic ranges, and the interspecies differences found by the authors, such a comparison cannot be considered valid. My opinion is that such a relationship should not be suggested based on the data provided in the paper.

Minor concerns

Abstract: the word „need” appears twice in the same sentence in lines 14 and 15. Please rephrase.

Line 295: the product code of the phenytoin standard should be provided considering that this information has been made available for all other standards in this part of the paper.

Figure 1: The coding with asterisks in the plot is not explained adequately in the caption. Please amend.

Author Response

(The authors gave the same response as above.)

Reviewer 3 Report

Comments and Suggestions for Authors

This manuscript presents a well-designed comparative analysis of PK/PD profiles of six clinically used ASMs using the MES model in both mice and rats. The inclusion of both plasma and brain EC50 values along with B/P ratios adds significant depth and translational relevance to preclinical assessment of ASMs. The study is well-written, methodologically sound, and addresses an important gap in the field regarding species-specific PK/PD variability. However, several areas can be further improved to enhance clarity, data interpretation, and translational applicability.

Major comments

  1. Clarification of brain sampling timing and distribution homogeneity: While the methodology section describes brain collection procedures post-seizure assessment, it would strengthen the rigor of the PK data if the authors clarified.

The precise timing between drug administration, seizure testing, and tissue collection.

Whether the brain samples were analyzed as whole-brain homogenates or specific regions.

As brain distribution can be heterogeneous, this information is essential for interpreting B/P ratios.

  1. Justification for dose ranges and clinical relevance: The selected dose ranges for each ASM vary widely between species (e.g., VPA: 200–750 mg/kg). It would be helpful to explicitly state the rationale behind these dosing regimens.

Were they based on known ED50 values, pilot studies, or clinical exposure equivalence?

Could the authors discuss how these doses correspond to human therapeutic exposures when normalized by allometric scaling or unbound concentrations?

  1. Statistical power and EC50 estimation robustness: While EC50 values were estimated using logistic regression, no confidence intervals or goodness-of-fit metrics are reported.

Please include EC50 confidence intervals and possibly R² or other fitting diagnostics in Supplementary Materials.

In addition, how were outliers or inconsistent responders handled in the logistic model?

  1. Translational relevance and recommendations for human studies: Figure 6 compares EC50 values with human therapeutic plasma concentrations. However, the authors should expand on.

How these findings can be used to improve first-in-human dosing strategies or dose selection for early clinical trials.

The relevance of free (unbound) drug concentrations, especially for highly protein-bound drugs like PHT and VPA.

Minor comments

  1. Abstract: Consider including key quantitative results (e.g., EC50 fold-differences between species) to immediately highlight the main findings.

  1. Figure legends: It would be clearer if each figure legend explicitly defined all abbreviations used and indicated whether bars represent SEM or SD.
  2. In Figure 1, there is a lack of specification for *** and ****.
  3. It is recommended that a footnote clearly explain how the “Therapeutic plasma concentration range” values ​​in Table 1 were derived.
  4. Figure 6: Please clearly state the source of "Therapeutic plasma concentration ranges observed in patients" for each drug.

Author Response

(The authors gave the same response as above.)

Round 2

Reviewer 1 Report

Comments and Suggestions for Authors

All comments were answered, and I have no more comment.

Author Response

Thank you for your thorough review and your suggestions to help improve our manuscript.

Reviewer 2 Report

Comments and Suggestions for Authors

Thank you for your replies to the issues raised in the review.

In my opinion the fact that a decision was made to write a paper based on some data that was available in the house does not justify the content. I maintain my opinion that the study design was poor. I also find any attempt of the authors to extrapolate these data to humans irresponsible in clear lack of any evidence that they could be correct in doing so.

Author Response

Dear Reviewer 2, 

In response to your concerns about relating our preclinical data to human exposure ranges without any proper PK modeling, we have now removed Figure 6 and any mentions of clinical data from the abstract, introduction, results and methods sections. All paragraphs in the discussion devoted to relating our MES data to clinical data have also been removed in their entirety with the associated references.

The MES assay as described in this work is a widely used test to assess the anti-seizure efficacy of drug candidates. Because it is important to us to provide the preclinical research community with a picture of how efficacious exposures from this commonly used single dose/single time point MES protocol relate to clinically reported effective exposures, and to honour reviewer 3's request to add unbound plasma concentrations to our comparison with clinical data, we kindly ask you to accept this information as Supplementary Table 1 with a short citation in one sentence in the discussion section (proposed edit highlighted in yellow in the revised manuscript). The caveats are already discussed and clearly laid out in this paragraph of the discussion. We strongly believe that simply stating these values without any extrapolations or interpretations puts our data in much needed context for preclinical researchers using the MES in drug discovery.

Kind regards,

Nina Weishaupt

Reviewer 3 Report

Comments and Suggestions for Authors

The manuscript is well present the related results.

Those results would be important to psycho-acting drugs' research.

Author Response

Thank you for your thorough review. We feel your suggestions have improved our manuscript.